# Methods of Sputum and Mucus Assessment for Muco-Obstructive Lung Diseases in 2022: Time to “Unplug” from Our Daily Routine!

**DOI:** 10.3390/cells11050812

**Published:** 2022-02-25

**Authors:** Jeremy Charriot, Mathilde Volpato, Aurélie Petit, Isabelle Vachier, Arnaud Bourdin

**Affiliations:** 1Department of Respiratory Diseases, Arnaud de Villeneuve Hospital, Montpellier University Hospital, CEDEX 5, 34295 Montpellier, France; m-volpato@chu-montpellier.fr (M.V.); aurelie.fort@inserm.fr (A.P.); isabelle.vachier@medbiomed.fr (I.V.); a-bourdin@chu-montpellier.fr (A.B.); 2PhyMedExp, University of Montpellier, INSERM U1046, CEDEX 5, 34295 Montpellier, France; 3Médecine Biologie Méditerranée, CEDEX 5, 34295 Montpellier, France

**Keywords:** asthma, chronic obstructive pulmonary disease, non-cystic fibrosis bronchiectasis, muco-obstructive lung diseases, sputum, mucus, mucins, rheology

## Abstract

Obstructive lung diseases, such as chronic obstructive pulmonary disease, asthma, or non-cystic fibrosis bronchiectasis, share some major pathophysiological features: small airway involvement, dysregulation of adaptive and innate pulmonary immune homeostasis, mucus hyperproduction, and/or hyperconcentration. Mucus regulation is particularly valuable from a therapeutic perspective given it contributes to airflow obstruction, symptom intensity, disease severity, and to some extent, disease prognosis in these diseases. It is therefore crucial to understand the mucus constitution of our patients, its behavior in a stable state and during exacerbation, and its regulatory mechanisms. These are all elements representing potential therapeutic targets, especially in the era of biologics. Here, we first briefly discuss the composition and characteristics of sputum. We focus on mucus and mucins, and then elaborate on the different sample collection procedures and how their quality is ensured. We then give an overview of the different direct analytical techniques available in both clinical routine and more experimental settings, giving their advantages and limitations. We also report on indirect mucus assessment procedures (questionnaires, high-resolution computed tomography scanning of the chest, lung function tests). Finally, we consider ways of integrating these techniques with current and future therapeutic options. Cystic fibrosis will not be discussed given its monogenic nature.

## 1. Sputum, Mucus, and Mucins in Healthy Subjects

### 1.1. Role and Components

Sputum is mucus coughed up from the lower airways. Mucus has a physiological role in humidifying the airway, acting as a physical and immunological barrier as well as participating in mucociliary transport. Mucus harbors unique biophysical properties, including viscoelasticity, an adjustable rheology, and a self-repairing capacity. Mucus is, therefore, an ideal medium for trapping and immobilizing external pathogens and toxins [1]. This important role explains the evolutionary conservation of mucus across various species from corals to humans [2]. 

Airway mucus is composed of water (98%), salt (0.9%), globular proteins (0.8%), and high molecular weight polymers (0.3%) in healthy subjects [3]. Considering the mucociliary apparatus as an entity, mucus is now represented by two gel phases: a mucus layer and a periciliary layer [4] (Figure 1). Mucus is mainly composed of mucins. These are large glycoproteins (approximately 400 kDa and 0.2–10 microns per polymer) containing regions rich in serine and threonine residues that can bind by O-Glycosylation to glycan chains.

There are several mucin genes encoding distinct mucins (MUCs), among which only seven are secreted. Only five of these secreted mucins can polymerize and thus participate in gel formation; three being secreted in the airways and all located at chromosome 11p15.5: MUC5AC, MUC5B, and MUC2 [5]. MUC5B is the dominant secretory mucin in the superficial epithelium (club cells and goblet cells) and submucosal glands, with distal airways being a major site of expression. MUC2 secretion is negligible [6].

### 1.2. Regulation of Airway Mucins 

It is important to distinguish between production (i.e., expression/transcription/synthesis) and secretion (exocytosis of mucin-containing granules) with regards to airway mucin regulation. Various pathways enhancing mucus production have been described [7]. One of the most studied is the EGFR/RAF/RAS/MEK/ERK pathway leading to mu5ac gene expression in response to several ligands such as lipopolysaccharides, TGF-alpha, amphiregulin, etc. [8,9]. Another pathway of interest, the IL13/STAT6/SPDEF pathway, is also involved in mucus production and is responsible for airway remodeling toward a mucosecretory phenotype (goblet cell hyperplasia and metaplasia) in obstructive lung diseases [10,11]. Beside this machinery of the surface epithelium, submucosal glands are of great interest since they are considered to produce nearly 90% of the total airway mucus [12]. The control of their secretion depends mainly on the cholinergic nervous system. The response to cholinergic stimulation is largely mediated by muscarinic M3 receptors [13], with water secretion being mediated by M1 receptors. These receptors are well known therapeutic targets. The functional alterations of these glands, for example, the loss of their ability to secrete fluids in response to any stimulus, or defects of cilia lining the glands’ ducts, are also pathophysiological pathways of interest [14]. 

The secretion of mature, polymerized mucins occurs continuously at low levels but can be amplified by many stimuli. In the basal state, mucin granule exocytosis is induced by low levels of activation of the purinergic P2Y2 and adenosine 3 receptors by paracrine-released extracellular ATP (and its metabolite, adenosine). Phospholipase C cleaves phosphatidylinositol 4,5-bisphosphate (PIP2) and generates di-acyl-glycerol (DAG) and inositol triphosphate (IP3). DAG activates Munc13, resulting in conversion of syntaxin into an open form; a soluble N-ethylmaleimide-sensitive factor attachment protein receptor (SNARE) complex with synaptosomal-associated protein 23 (SNAP-23) and vesicle-associated membrane protein (VAMP). This complex brings the plasma and granule membranes into close proximity. IP3 induces calcium release from the endoplasmic reticulum to the cytoplasm via activating IP3 receptors, then activating the synaptotagmin-mediated coiled-coil conformation of the SNARE complex, resulting in membrane fusion and mucin release [15,16].

Given these data, we can easily imagine that preserving a MUC5AC/MUC5B ratio is crucial for preventing the development of obstructive lung diseases [17].

## 2. Mucus and Mucins in Muco-Obstructive Lung Diseases

Richard C. Boucher and his team from the University of North Carolina—Chapel Hill put forward the term “muco-obstructive lung diseases” to encompass lung diseases with the following common pathophysiological features: small airway involvement with mucus plugging, dysregulation of both mucin homeostasis (hyperconcentration) and mucin biophysical properties (increased mucus viscosity and elasticity), and airflow obstruction [18]. We move on now to discuss asthma, harboring all these characteristics, despite specific pathological pathways of innate and adaptive immune responses being involved.

### 2.1. Chronic Obstructive Pulmonary Disease

Chronic obstructive pulmonary disease (COPD) is characterized by respiratory symptoms and airflow obstruction [19]. Small airways are affected earlier [20] partly due to mucus hypersecretion. This abnormality is correlated with the severity of airflow obstruction and mortality [21]. Kesimer et al. showed in patients with severe COPD from the SPIROMICS (Subpopulations and Intermediate Outcomes Measures in COPD Study) cohort that absolute concentrations of MUC5B and MUC5AC were 10-times higher than those in healthy subjects. They also showed that total mucin concentration can predict chronic bronchitis [22]. The same team later demonstrated in the same population that the MU5AC concentration in induced sputum had a greater correlation than the MUC5B concentration with COPD features (FEV1 (Forced Expiratory Volume in the 1st second), exacerbation rate, hyperinflation) [23]. Several mechanisms have been proposed to underlie these observations, including: (1) an obvious effect of past or current cigarette smoke exposure on airway remodeling (goblet cell hyperplasia and metaplasia) [21,24], on mucus production/secretion (particularly via the epidermal growth factor receptor (EGFR) [9,25,26]), and on mucus hydration [27,28,29]. (2) A qualitative alteration of the mucin network, enhanced in the contexts of chronic infection and/or acute exacerbations [1,30,31,32]. (3) An impairment in mucociliary clearance that could deteriorate in cases with ciliary motility dysfunction [33]. 

### 2.2. Asthma

Mucus plugs have long been described in asthma, especially in fatal cases [34]. In severe asthma, mucus plugs are linked to the severity of the airflow obstruction and with sputum eosinophilia [35]. Mucus and, notably, MUC5AC hyperproduction have also been observed in patients with mild to moderate asthma, whereas muc5b gene expression is reduced [36], suggesting a multitude of pathophysiological mechanisms leading to such abnormalities [37,38]. Although distinct endotypes (Th2-High and Th2-Low) have been described in asthma [39], we focus here on the importance of the IL-13/SPDEF/STAT6 pathway (interleukin-13/SAM pointed domain-containing ETS transcription factor/signal transducer and activator of transcription 6). This pathway drives airway remodeling toward a different mucosecretory phenotype, goblet cell metaplasia and hyperplasia [40,41], muc5ac overexpression [42], and presumably a reduced production of mucin MUC5B. The latter completely destabilizes the MUC5AC/MUC5B ratio, the increase of which is linked to the Th2-High endotype [43]. This is consistent with the observation that reduced MUC5B in mucus can favor eosinophil survival [44]. In parallel, another study showed that a MUC5AC-rich mucin network is more strongly tethered to the epithelium, impairing mucociliary clearance [45]. These data highlight the fact that mucus quality is as important as quantity with respect to the underlying pathophysiological causes. Interestingly, bronchoconstriction alone is also able to induce a secretory epithelial phenotype and mucus hypersecretion [46], suggesting cross-talk between airway smooth muscle cells (ASMc) and secretory cells that could act via the epithelium-derived cytokine CCL20 and its receptor CCR6 [47,48].

### 2.3. Non-Cystic Fibrosis Bronchiectasis

Proposing a common pathophysiological cause for mucus overproduction/over secretion in non-cystic fibrosis (CF) bronchiectasis, diagnosed by chest high-resolution computed tomography (HRCT), is a daunting challenge due to the heterogeneity of this underlying entity and multitude of etiologies [49]. The first histopathological studies revealed severe small airway disease, bronchiolectasis, and mucus plugs [50]. In addition, although chronic bronchitis and exacerbations are a hallmark of non-CF bronchiectasis, they are not systematically reported. Data on mucus and mucin regulation are minimal and often extrapolated from CF despite it being a monogenic disease. The main alterations observed in the sputum of patients are the hyperconcentration of MUC5AC and MUC5B mucins, as well as an increase in their viscous and elastic properties. Nonetheless, no increases in mucin gene expression have been found [51], suggesting that mucus is in a dehydrated state triggered potentially by local hypoxic inhibitions of cystic fibrosis transmembrane conductance regulator (CFTR) activity [52], or by fluid absorption enhanced by human neutrophil elastase (HNE) activity [53]. Noteworthy, 50% of patients with an incident diagnosis of bronchiectasis are found to have at least one CFTR mutation that does not necessarily lead to diagnosis of CF [54]. Ultimately, the effect of chronic infection and exacerbation must be commented on, even if there are few in vivo studies specifically addressing this issue in non-CF bronchiectasis. Considering that in vitro and ex vivo models of *Pseudomonas aeruginosa* (PA) infection show that mucus downregulates the basal levels of bacterial virulence genes [55,56], we can speculate that alteration of mucin homeostasis and the mucin network in bronchiectasis contributes to mucociliary impairment and increased susceptibility to infection.

Overall, mucus and mucins are determinants of the underlying pathophysiology of obstructive airway diseases. However, reliable and relevant measurement of these parameters in humans remains a challenge.

## 3. Sample Collection

### 3.1. Flexible Bronchoscopy

Flexible bronchial fibroscopy (also called bronchoscopy) is the mucus collection technique that immediately springs to the mind to any respiratory physician. It allows for a subjective, non-quantitative, macroscopic study of mucus secretion and bronchial mucosa (dilated bronchial mucus glands, atrophy) [57]. Bronchoscopy can also be used to aspirate mucus and even mucous plugs, and/or to perform a bronchoalveolar lavage (BAL). BAL fluid composition reflects that of non-contaminated mucus in small airways. However, retrieval of BAL fluid is variable, especially in patients with very severe small airway involvement [58]. Bronchoscopy also allows for bronchial biopsies or epithelial brushings to be taken for airway remodeling assessment [59]. Despite being robust for airway remodeling assessment, performing the examination itself can result in the modification of several parameters that can affect mucin network composition or quality; for example, lidocaine is frequently instilled into the airway and can reduce mucus production [60,61]. Saline solution is used for BAL and can change the rheological properties of mucus [62]. Finally, coughing, physical stress, or mechanical stimulation induced by the fiberscope can favor mucus secretion through the activation of the cholinergic system [63]. The drugs used for anesthesia (when general anesthesia is preferred) can effect mucociliary clearance [64]. The required medical setting, as well as the invasive nature of the fibroscopy and related morbidity for the most severe patients remain the main limitations of this examination.

### 3.2. Sputum

Sputum appears the simplest way to collect mucus via coughing. No medical supervision is needed for spontaneous sputum sampling. In contrast, “induced sputum” requires serial and standardized spirometric testing (Figure 2). Clinical evaluation is also required as nebulized hypertonic saline can lead to bronchial hyperresponsiveness [65,66]. Standardized processing procedures (purification, centrifugation, no freezing) are also crucial to obtain reliable data [67]. The quality of these samples is readily assured by cytological examination (<25% squamous cells [68]) and the presence of alpha-amylase activity in the sample [22] (even though no consensual threshold has been defined). Comparative studies of spontaneous versus induced methods have shown similarities in total cell counts, but more viable cells are observed in induced sputum samples [69]. Moreover, a correction factor needs to be applied when assessing the percentage of solids and mucins in mucus due to the use of hypertonic saline [70]. Another potential data analysis pitfall is that hypertonic solution also modifies sputum rheology, but the resulting variations follow the same trend as those for all other studied pathologies [71,72]. Although relevant to understanding the pathophysiology of obstructive lung diseases, studies based on spontaneous sputum methods are restricted to subjects able to expel, with induced sputum, thus, not being a feasible option in patients with severe airflow obstruction. These factors could introduce selection bias in such experiments.

## 4. Direct Assessment of Human Airway Mucus and Mucins 

Bearing in mind the advantages and disadvantages, as well as the potential biases and difficulties associated with sampling human airway mucus, the “direct analysis” of mucus remains a valuable and varied source of information. We cover direct assessment in detail below (Figure 3; Table 1).

### 4.1. Macroscopic Studies of Sputum

Patients are the first to notice changes in their sputum. In his now famous 1987 publication, Anthonisen defined COPD exacerbation by increased purulence and sputum production [73]. The introduction of antibiotics was thus approved in this case. The same definition is still used in the most recent guidelines [19]. The limitations of this assessment are intuitive: the definition of purulence is inaccurate (from uncolored to yellow-green) and the link with bacterial infection of the lower respiratory tract is controversial [74,75]. Indeed, the presence of pathogens in sputum does not necessarily imply a need for antibiotics [76]. Moreover, this “technique”, originating from the context of COPD, can hardly be extrapolated to asthma—where sputum color may be linked to neutrophilia—and does not replace sputum cytology [77,78].

### 4.2. Cytology

Performing total cell counts is a useful routine technique for assessing sputum quality [79] and for classifying airway inflammation. This is especially the case in asthma [80] where sputum samples with: (1) normal neutrophil (<61%) and eosinophil (<1.9%) counts are considered paucigranulocytic; (2) normal neutrophil counts and raised eosinophil counts (≥2%) are considered eosinophilic; (3) raised neutrophil counts (≥61%) and normal eosinophil counts are considered neutrophilic; (4) both raised neutrophil and eosinophil counts are considered mixed granulocytic. Cytology is required for understanding mucus regulation, i.e., neutrophilia has been linked to mucus hypersecretion through an increase in human neutrophil elastase (HNE) release [81]. Furthermore, sputum hypereosinophilia has been shown to modify mucus and favor mucus plugging, notably through the formation of Charcot–Leyden crystals in obstructive diseases such as asthma [35,82]. Cytology protocols are now well standardized and their limitations lie essentially with sampling quality [79].

### 4.3. Microbiology 

We previously described the influence of the microbiota and acute or chronic infection on airway mucus regulation. The current gold standard for sputum microbiological examination is based on smear microscopy (bacteria, mycobacteria, Aspergillus), their subsequent identification after staining and/or biochemical tests (motility, McFarland standard, fluid thioglycollate medium, catalase, and oxidase tests), and then microbial culturing in different media. Antimicrobial susceptibility is then tested when possible [83]. There are numerous quality checks for each step of this technique, yet the low rate for diagnosing lower respiratory tract infection (LRTI) found in the literature raises doubts; ranging from 13% for this non-invasive techniques versus up to 56% for more invasive sampling (BAL fluid) [84,85]. The main limitation reported is again sample quality, which can be contaminated by pathogens from the upper airways or pathogens non-representative of the distal airway microenvironment.

The development of the Nucleic Acid Amplification Test (NAAT) procedure allows us to identify an ever-increasing range of pathogens, including viruses, with increased sensitivity and very high specificity [86]. This is particularly valuable during acute exacerbations of obstructive lung diseases induced by viruses, notably asthma with viruses accounting for over half of even adult cases [87]. Although beneficial for reducing unnecessary exposure to antibiotics, the increased use of multiplex molecular assays for infectious respiratory diseases [88] or stable obstructive lung diseases also raises new concerns given their high sensitivity and specificity for pathogen identification. Fine-tuned pathogen identification will likely lead us to rethink the notions of pathogen colonization, chronic infection, and pathogenicity.

### 4.4. Histological Staining and Immunostaining

Airway tissue samples (bronchial biopsies or brushings, lung explants) are routinely analyzed by histological stains (periodic acid-Schiff (PAS), Alcian blue (AB)) that are simple and widespread, standardized techniques [89]. Fluorescent lectins can be used for semi-quantitative analysis by fluorescence intensity measurement and they are inexpensive, although rather reserved for experimental settings [90]. It should be noted that non-specific labeling can only analyze the mucus and periciliary layers combined and not mucins alone [91]. Staining with antibodies (anti-MUC5AC/anti-MUC5B) is more specific and allows for quantification of distinct mucin subtypes and cell differentiation, as well as mucociliary clearance assessment [4,92]. These techniques may also be coupled with rheological assessments [51], for instance with fluorescent probes or dyes [93]. In a future increasingly oriented towards precision medicine and treatable traits, combined assessment allows for precise endotyping and a reasonably accurate measure of the effects of a therapeutic intervention [94]. The limitations of these analyses are inherent to the technology, environmental conditions, and parameters employed [95,96]. 

### 4.5. Molecular Assays 

Muc5AC and Muc5B gene expression in obstructive lung diseases have been dissected in numerous studies, and notably in cigarette smoke exposure models [9,22,26,97,98]. Quantitative RT-PCR, northern blot, or in situ hybridization are inexpensive methods allowing for accurate evaluation of mucin mRNA levels in various conditions [6]. However, these analyses must be carefully considered when it comes to in vivo extrapolation. Firstly, most studies are performed in human bronchial epithelial cells cultured at air-liquid interface [99]. This model is highly valuable given its ability to reproduce pathological airway features [100,101], but the model itself harbors some limitations that we develop in a later section. Secondly, mucin production analyzed by mRNA levels may not be the most suitable way to investigate muco-obstructive lung diseases; mRNA levels do not always correspond to mucin protein levels and post-transcriptional modifications are not detected. Indeed, an increase in intracellular mucin-containing granules, mucin hyperconcentration [51], dysregulation of epithelial fluid transfer [52], or qualitative changes in the mucin network [45] are likely more relevant observations of mucin production in muco-obstructive lung diseases. More recent techniques, such as chromatin immunoprecipitation (ChIP)—the study of DNA-protein/transcription factor binding interactions [102,103]—or single-cell RNA-sequencing [104,105,106,107]—extremely relevant for understanding cells populations and lineages in heterogeneous samples—give us a dynamic overview of the airway epithelium and its mucosecretory cells. We do not detail the genetic approaches essentially based on experimental animal models already described elsewhere [91]. Moreover, their use is seldom transposable to daily clinical routine.

### 4.6. Semi-Quantitative and Quantitative Assessments of Mucin Proteins 

Enzyme-linked immunosorbent assay (ELISA) or antibody detection-based Western blotting techniques are the most widely used for detecting mucin proteins [108,109,110]. Sandwich ELISA appears preferable as it offers a greater sensitivity and specificity than absorption ELISA [111]. Sputum or BAL volume is a major limitation for these techniques given that the sampling method can directly affect mucin concentrations. Another pitfall is the potential alteration of mucin epitope which can lead to underestimation of mucin measurements [51]. For instance, Henderson et al. showed that MUC5B concentrations in CF sputum is systematically underestimated or undetected when immunoblotting in samples containing PA [112]. The authors suggest that this lack in detection could be explained by proteolytic cleavage of MUC5B at antibody recognition sites induced by PA. The authors propose performing mucus analysis by mass spectrometry to overcome this issue. Mass spectrometry is a highly sensitive proteomic technique for the identification of protein types present along with their relative abundances in human mucus; from which comes the notion of “sputome” [113,114,115]. Conversely, percent solids (wt%) represent a much simpler and affordable method that is proving correlated with total mucin and DNA concentration in non-CF bronchiectasis mucus [51,116]. The technique does, however, give no information on mucus composition. The dot-blot assay is also a simple technique for detecting glycoproteins but severely lacks in sensitivity and specificity [117].

### 4.7. Biophysical Properties and Rheology of Human Mucus

The physical behavior of the mucin network is highly influenced by its ratio of MUC5A/MUC5B mucins [43], the structural and conformational changes of each of these mucins (O-glycosylation, degree of sulfation, ionic charges) [110,118], and, consequently, interactions between the mucin network and other proteins (DNA, bacteria, etc.) or solvents/fluids [119]. In addition, rheology—the science of fluid flow and deformation—is considered valuable in the study of mucus and mucociliary clearance. Like any physical sciences, there are several methodological options for assessing rheology depending on the level of accuracy desired [120]. Mucus nanorheology can be performed with fluorescence recovery after photobleaching (FRAP) assays [121] using probes generally with a 5 nm diameter. This technique can assess the properties of solvents and down to the smallest protein component of mucus. Investigating the differences in mucus with varying concentrations of solid has proven interesting [122]. Microrheology is suitable for studying the local rheology of mucus considered as a single layer [123] at the epithelial cilia scale. One-μm particles are generally used to ensure that the study probes are larger than the correlation length (mesh size) of mucus. This method is, therefore, particularly suited to the analysis of mucus secreted by human bronchial epithelial cells cultured ex vivo at an air-liquid interface. With the use of micro-probes, Jory et al. showed that mucus gradually varies in rheological response, from an elastic behavior close to the epithelium to a viscous behavior further away [124]. Lastly, a fraction of a mucus sample can be directly assessed. This is the field of macrorheology. Macrorheology can be performed by specific devices including the cone-and-plate rheometer, capillary viscometer, or filancemeter. They enable the estimation of viscosity and elasticity under various physiological or pathological conditions [120], as well as indicating the effect of a therapeutic intervention on mucus rheology, such as hypertonic saline [72,125]. This perspective also offers the possibility of focusing on mucus adhesion and cohesion in cough clearance [126]. Interestingly, Patarin et al. demonstrated that sputum rheology assessed by a “miniaturized” cone-and-plate rheometer could be a useful biomarker for distinguishing various muco-obstructive lung diseases [71]. Additionally, our team has shown that results from rapid on-site sputum rheology assessment not only correlate with mucin concentrations analyzed by mass spectrometry, but the results can also predict sputum eosinophilia irrespective of the underlying disease (*clinicaltrial NCT04081740, manuscript submitted*). Such methods remain particularly relevant in the biologic era.

### 4.8. Ex Vivo Models

This paragraph deliberately targets the frontier between direct and indirect mucus assessments. The most widely used models for studying the mucus are indeed ex vivo, with the most relevant model being reconstituted airway epithelia from patient bronchial biopsies. These epithelia are cultured at air-liquid interface in order to mimic in vivo conditions as much as possible. These cultures retain the “pathological” phenotype of the patients from which they derive (percentage of ciliated, club, or goblet cells) [127]. Ex vivo airway epithelial cell models thus provide a biologically representative model for investigating airway diseases [100,101]. Numerous current or potential drugs can be tested and molecular mechanisms elucidated via this model [128,129]. However, experienced laboratory team members are crucial for cell culture success as contaminations are frequent [130]. Moreover, whether it be an advantage or disadvantage, the reconstituted ex vivo epithelia are free from interactions with other neighboring functional “compartments” (smooth muscle cells, vessels, collagen, and fibroblasts, etc.), and this is despite the fact that co-cultures can be established [131]. This may obviously limit the ability to extrapolate some findings. The model may also not completely represent the pathophysiological mechanisms involved in distal airways given biopsies are taken from proximal bronchial airways. Three-dimensional organoids mimicking the major characteristics of their in vivo counterpart organs could overcome this limitation in the future [132,133,134].

## 5. Indirect Assessment of Human Mucus Dysregulation

Mucus dysregulation can be indirectly investigated via the resulting clinical and paraclinical consequences that we briefly develop thereafter (Figure 3; Table 1).

### 5.1. Respiratory Symptoms

The most clinically-meaningful symptom that could be due to mucus hyperproduction in muco-obstructive lung diseases is chronic bronchitis. Chronic bronchitis is defined by a cough and sputum for at least three months a year and for two consecutive years [19]. Chronic bronchitis remains within the diagnostic criteria for COPD despite it being reported among 7.4–74% of patients with COPD [135,136]. This varied prevalence of chronic bronchitis in COPD is study dependent, but does highlight the subjective nature of chronic bronchitis. Nevertheless, chronic bronchitis in COPD has been found related to exacerbations and mortality [137]. Validated questionnaires have been developed to obtain semi-quantitative scales that can be used in clinical trials and daily clinical practice. The Cough and Sputum Assessment Questionnaire (CASA-Q), mostly used in COPD and asthma, is very specific to chronic bronchitis and has the advantage of assessing both symptom intensity and their effects [138,139,140]. Strikingly, Alagha et al. showed that chronic bronchitis assessed by CASA-Q was not correlated with goblet cell hyperplasia in asthma, calling into question the reliability of such clinical evaluation.

### 5.2. Imaging: High-Resolution and Micro-Computed Tomography Scanning

Chest HRCT alone cannot confirm a diagnosis of obstructive lung disease (with the exception of bronchiectasis which is diagnosed by imaging). Chest HRCT can, however, reveal some manifestations of the disease: air trapping [141], mucus plugs [35,142], atelectasis, bronchiolitis (tree-in-bud), airway wall thickness [143,144]. Spirometer-triggered HRCT allows for structure-function analysis of small airways, especially when performed after a methacholine challenge [145,146,147]. Dunican et al. developed a bronchopulmonary segment-based scoring system to quantify mucus plugs on HRCT scans. The authors found a high mucus score was associated with: (1) lower FEV1 and marked increases in sputum eosinophils in patients with asthma, and (2) lung function outcomes in smokers with limited emphysema [35,148]. However, one may object that this technique remains imprecise for characterizing the real nature of bronchial occlusion. Indeed, distinguishing between differences in mucus density, increased wall thickness, or smooth muscle hyperplasia remains unfeasible. Micro-CT is particularly valuable for the understanding of the development of small airway diseases directly from human lung samples [21,149,150]. Unfortunately, this very anatomical approach has not yet been applied to the study of mucus in humans, but some early work has been initiated in animals to assess mucociliary transport [151].

## 6. Conclusions

Mucus dysregulation could be considered as the be-all and end-all of certain obstructive lung diseases, such as asthma, COPD, or non-CF bronchiectasis. Despite the wide range of available methods for assessing mucus production and secretion, as well as the biophysical and biochemical properties of mucus, daily clinical practice has relied for years now on clinical evaluation, chest HRCT, cytology, and microbiology. These examinations have an extremely variable added value when considering endotyping and provide random benefits to patients. Our growing knowledge of the pathophysiological mechanisms underlying these diseases is opening up new perspectives. Indeed, improving current techniques and developing new models to collect and analyze human mucus samples and study the in vivo pathophysiological mechanisms are perquisites for understanding the intra-individual differences in mucus dynamics and mucus interactions with the (micro)environment. Not every technique described in this review may be widely used due to the expertise and time-processing required. Given the miniaturization of devices, mucus macrorheology assessment now appears to be amongst the more readily accessible biomarkers for current medicine, with tangible connections between past, current, and future therapeutics and clinical outcomes. Indeed, mucolytics (i.e., hypertonic saline and rhDNAse) are able to influence mucus hydration [71,125], whereas macrolides (azithromycin, erythromycin) inhibit mucus secretion and have proved to reduce exacerbations in obstructive lung diseases [152,153,154]. In the wake of monoclonal antibodies within the asthma context (anti-IL5/5R, anti-IL-4/13R, anti-TSLP) [155,156,157,158], some of which directly affect airway remodeling or even inhibit mucus secretion, the implementation of “basic science” methods as a point-of-care or simply to guide therapy would be of great value (COPD CaRhe—clinicaltrial.gov, accessed on 21 December 2021, NCT04339270). Finally, in order to better understand the complexity of mucus regulation, it is likely that a multimodal approach will enable us to progress towards what we call “tailored medicine” and may even allow for early disease detection.

## Figures and Tables

**Figure 1 cells-11-00812-f001:**
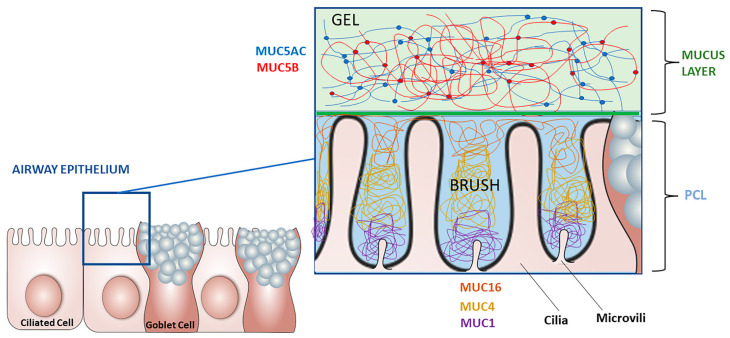
Two-gels model of mucus transport, inspired from Richard C. Boucher—The New England Journal of Medicine. (PCL: periciliary layer).

**Figure 2 cells-11-00812-f002:**
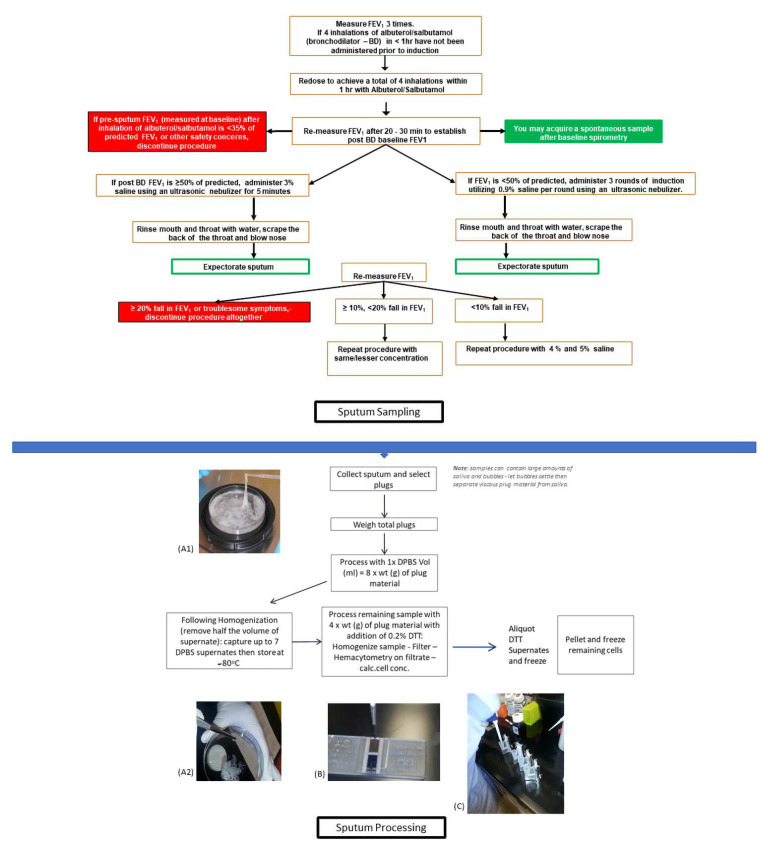
Proposed protocol for Induced Sputum Sampling and Processing (calc.cell.conc = calculate cell concentration, FEV1 = Forced expiratory Volume in the 1st second, DPBS = Dulbecco’s phos-phate-buffered saline, DTT = dithiothreitol, wt = Weight total). (**A1**) Sputum is first collected in a petri dish (**A2**) Plugs are selected (**B**) Total cells and cell viability are calculated from the sample filtrate (**C**) Cytospin set up (pipette delivery of sample into cytofunnel).

**Figure 3 cells-11-00812-f003:**
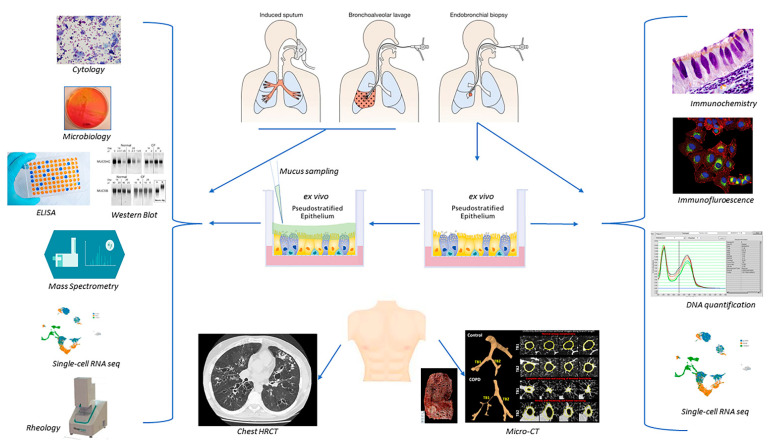
Summary of the main current direct and indirect techniques for assessing airway mucus and mucins. ELISA: enzyme-linked immunosorbent assay; HRCT: high-resolution computed tomography; RNA: ribonucleic acid.

**Table 1 cells-11-00812-t001:** Strengths and limitations of the main current direct and indirect techniques for assessing airway mucus and mucins.

	Production	Secretion	Biophysicial Behaviour	Strengths	Limitations
** *Total cell count (sputum/BAL fluid)* **	+/−	−	**+** *(indirect link)*	Simple, unexpensive, performed routinely in health facilities	Information provided about mucus is limited
** *Microbiology* **	+/− *(indirect link)*	+/− *(Indirect link)*	+/− *(indirect link)*	Simple, unexpensive, performed routinely in health facilities	No clear direct correlation *in vivo* bacterial load and mucus production/secretion
** *ELISA (mucins)* **	**++** *(cell lysates)*	**++**	−	Quantitative assay, can be used *in vivo* Rapid and simple measurement of intra- or extra-cellular mucins	Caution needed with sample processing and epitope integrity, or homologous regions
** *Western Blot (mucins)* **	**++** *(cell lysates)*	**++**	−	Rapid and simple measurement of intra- or extra-cellular mucins	Semi-quantitative Caution needed with sample processing and verification of specificity Requires denaturation of mucins for agarose gel electrophoresis
** *Immunohistochemistry and Immunofluorescence* ** ** *(secretory cells, mucus)* **	**+** *(intracellular mucins)*	**+**	−	Spatial localization of mucins in airway and/or in secretory cells, co-localization with other components	Qualitative or semi-quantitative assessment. Time-consuming Scoring system needs blinded individuals
** *Mass Spectrometry* **	**++**	**++**	−	Accurate quantitative assay Very high specificity	Not routinely performed expensive, Time-consuming
** *Quantitative RT-PCR* ** ** *(mucins mRNA)* **	**+++**	−	−	Simple, unexpensive Specific quantitative information on mucin expression at the mRNA level	No detection of post-transcriptional modifications
** *Single-cell RNA seq* **	**+** *(intracellular mechanisms)*	**+** *(intracellular mechanisms)*	−	Dynamic overview of the intracellular mucus-secreting/producing machinery	No quantitative assessment of mucin production and secretion Expensive and time-consuming
** *Rheology* **	−	**+**/− *(indirect link)*	**+++** *(viscoelastic properties)*	Can be easily performed with *in vivo* samples Relationship with clinical phenotyping	The yield of sputum collection is variable Caution needed with sample processing and quality check (salivary contamination)
** *Ex vivo models* **	**++**	**++**	**+**	Chronic airway disease phenotype is maintened Measurement of exposure to drugs or toxins is feasible	High expertise needed for cell culture, mucus sampling can be difficult (PBS washes) Time consuming
** *HRCT* **	−	**+** **(indirect scoring)**	−	Routinely performed, unexpensive *In vivo* endotyping of chronic airway disease	No specific information on mucus production or secretion
**Micro-CT**	−	**+** **(indirect observation)**	−	High-quality *ex vivo* imaging of small airway diseases	expensive Invasive method (surgical lung biopsy or lung explant)

## Data Availability

Not applicable.

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
