# Peer review of "Methods of Sputum and Mucus Assessment for Muco-Obstructive Lung Diseases in 2022: Time to “Unplug” from Our Daily Routine!"

_cells, 2022, doi:10.3390/cells11050812_

Round 1
Reviewer 1 Report
The authors summarize the presently available methods and technics utilized in sputum and mucus analysis regarding COPD and asthma. Also, they provide a brief overview concerning mucus synthesis, intracellular transport, and exocytosis. The manuscript encompasses a copious amount of knowledge, which is this review's strength and simultaneously its weakness. The immense knowledge listed in this work almost touches every aspect that needs to be known in this field; consequently, it is hard to follow.
Major concern:
- Airway submucosal glands are needed to be discussed because they are one of the primary sources of mucus production in the human airways.
- Some statements must be clarified from line 74 to line 82. For example, 1) Phospholipase C cleaves phosphatidylinositol 4,5-bisphosphate (PIP2) and generates DAG and IP3. 2) IP3 induces calcium release from the endoplasmic reticulum to the cytoplasm via activating IP3 receptors.
Author Response
Dear Reviewers,
We’d like to warmly thank you for spending time to assess our manuscript and formulate pertinent and accurate comments.
Please find the tracked-changes version of the revised manuscript.
Additionally, please find below our point-by-point answers.
Should you have any other communication needs please do not hesitate to contact me back.
Best regards.
Dr J. Charriot for all the authors.
Reviewer 1:
The authors summarize the presently available methods and technics utilized in sputum and mucus analysis regarding COPD and asthma. Also, they provide a brief overview concerning mucus synthesis, intracellular transport, and exocytosis. The manuscript encompasses a copious amount of knowledge, which is this review's strength and simultaneously its weakness. The immense knowledge listed in this work almost touches every aspect that needs to be known in this field; consequently, it is hard to follow.
Major concern:
- Airway submucosal glands are needed to be discussed because they are one of the primary sources of mucus production in the human airways.
- Some statements must be clarified from line 74 to line 82. For example, 1) Phospholipase C cleaves phosphatidylinositol 4,5-bisphosphate (PIP2) and generates DAG and IP3. 2) IP3 induces calcium release from the endoplasmic reticulum to the cytoplasm via activating IP3 receptors.
Thank you for your comment. We have added a paragraph about submucosal glands (L.70-77), which is indeed often overlooked in protocols.
According to your remarks, we have also changed statements about mucins exocytosis.
Reviewer 2 Report
This is an interesting review about mucus and sputum assessment.
In my opinion, authors should add a separate chapter with possibly a nice figure or table, describing in detail the method of inducing sputum and sputum processing methods (entire vs selected plug, supernatant and pellet, etc.) with technical considerations.
Author Response
Dear Reviewers,
We’d like to warmly thank you for spending time to assess our manuscript and formulate pertinent and accurate comments.
Please find the tracked-changes version of the revised manuscript.
Additionally, please find below our point-by-point answers.
Should you have any other communication needs please do not hesitate to contact me back.
Best regards.
Dr J. Charriot for all the authors.
- This is an interesting review about mucus and sputum assessment.
- In my opinion, authors should add a separate chapter with possibly a nice figure or table, describing in detail the method of inducing sputum and sputum processing methods (entire vs selected plug, supernatant and pellet, etc.) with technical considerations.
Thank you for your review. We now propose in the revised manuscript, in the “Sputum” chapter a figure describing the method for induced sputum from sampling and processing.
Round 2
Reviewer 1 Report
The manuscript was sufficiently improved, and correction has been made according to the suggestions. An additional figure (Figure 2) is added to the manuscript that describes the sputum sampling and processing. This figure provides a more visual description of the process and aids comprehension.
Reviewer 2 Report
For me now it's ok to accept.